# Analyzing COVID-19 disinformation on Twitter using the hashtags #scamdemic and #plandemic: Retrospective study

**Heather D. Lanier**[ID]*, **Marlon I. Diaz, Sameh N. Saleh, Christoph U. Lehmann**[ID]**, Richard J. Medford**[ID]

Clinical Informatics Center, UT Southwestern Medical Center, Dallas, Texas, United States of America

* heather.lanier@utsouthwestern.edu

**Data Availability Statement:** All currently active tweets are available on Twitter.com. All Tweet IDs are within the Supporting Information files. Per

## Abstract

### Introduction

The use of social media during the COVID-19 pandemic has led to an "infodemic" of mis- and disinformation with potentially grave consequences. To explore means of counteracting disinformation, we analyzed tweets containing the hashtags #Scamdemic and #Plandemic.

### Methods

Using a Twitter scraping tool called twint, we collected 419,269 English-language tweets that contained "#Scamdemic" or "#Plandemic" posted in 2020. Using the Twitter application-programming interface, we extracted the same tweets (by tweet ID) with additional user metadata. We explored descriptive statistics of tweets including their content and user profiles, analyzed sentiments and emotions, performed topic modeling, and determined tweet availability in both datasets.

### Results

After removal of retweets, replies, non-English tweets, or duplicate tweets, 40,081 users tweeted 227,067 times using our selected hashtags. The mean weekly sentiment was overall negative for both hashtags. One in five users who used these hashtags were suspended by Twitter by January 2021. Suspended accounts had an average of 610 followers and an average of 6.7 tweets per user, while active users had an average of 472 followers and an average of 5.4 tweets per user. The most frequent tweet topic was "Complaints against mandates introduced during the pandemic" (79,670 tweets), which included complaints against masks, social distancing, and closures.

### Discussion

While social media has democratized speech, it also permits users to disseminate potentially unverified or misleading information that endangers people's lives and public health interventions. Characterizing tweets and users that use hashtags associated with COVID-19 pandemic denial allowed us to understand the extent of misinformation. With the

Twitter's guidelines to have a developer account, the full tweets are unable to be shared, only the Tweet IDs. If these guidelines are broken, the developer account is suspended and/or terminated.

**Funding:** The author(s) received no specific funding for this work.

**Competing interests:** The authors have declared that no competing interests exist.

preponderance of inaccessible original tweets, we concluded that posters were in denial of the COVID-19 pandemic and sought to disperse related mis- or disinformation resulting in suspension.

## Conclusion

Leveraging 227,067 tweets with the hashtags #scamdemic and #plandemic in 2020, we were able to elucidate important trends in public disinformation about the COVID-19 vaccine.

## Introduction

In 2021, almost four billion people were users of social media with the average user managing more than eight accounts on various social media platforms [1]. One such platform is Twitter, which has over 199 million daily monetizable active users and allows individuals to post, repost, like, and comment on 'tweets' of up to 280 characters that may include links, videos, or images. The vast majority of the posts are public [2].

Social Media can be the source of several types of false information: Misinformation, Disinformation, and Malinformation. Misinformation is false information not intended to harm. Disinformation is also false but carries the intent to harm. Malinformation represents genuine information intended to harm and may include leaks, harassment, and hate speech [3]. For our Twitter analysis, we selected two hashtags that represent mis- and disinformation (#plandemic and #scamdemic) to analyze the effect of false information.

The analysis of Twitter content has been used previously within the public health realm to understand public sentiment and gauge opinion on topics such as diabetes, the Affordable Care Act [4], social distancing [5], influenza [6], and measles [7]. Twitter may serve as a robust medium to better understand wide-scale, organic public perception about the COVID-19 pandemic [3,8,9]. Social media use during the COVID-19 pandemic has led to an "infodemic" generating mis- and disinformation with potentially grave consequences [10,11]. Starting in 2021, Twitter began applying labels to tweets that potentially contained misleading information about COVID-19. Twitter applied this new labeling policy to limit tweet visibility and spread of mis- and disinformation. Twitter mandated tweet removal across 11.5 million accounts and permanently suspended over 150,000 accounts for distributing misinformation [2,12].

The hashtags #scamdemic and #plandemic, which imply that the pandemic is a conspiracy, are frequently associated with intentional disinformation; however, tweets with these hashtags have not been examined to explore the scope of disinformation [13]. Understanding the extent and impact of false information is important for officials and public health agencies to predict population behavior including the potential uptake of vaccines and non-pharmaceutical measures such as masking and social distancing. Our hypothesis was that analysis of tweets associated with these hashtags would provide valuable insight about disinformation and the public's beliefs around the COVID-19 pandemic and would aid in developing targeted public health interventions.

## Methods

### Data collection and processing

On January 3, 2021, using the Twitter scraping tool Twint, we collected English-language tweets that contained the hashtags "#scamdemic" or "#plandemic" and were posted between

January 1 and December 31, 2020. Subsequently on January 15, 2021, we used the Twitter application programming interface (API) to extract the same tweets (using the corresponding tweet IDs) to collect additional relevant metadata. We provided descriptive statistics for tweets including user profiles and tweet content and determined tweet availability in both datasets based on Twitter API status codes (User has been suspended or No status found with that User ID). We used Python version 3.9.1 software (Python Software Foundation, Wilmington, DE) for all data processing and analyses. Institutional review board approval was not required because this study used only publicly available data.

### Sentiment & subjectivity and emotion analysis

To perform sentiment analysis for tweets, we tokenized them and cleaned and transformed tokens into their root form through natural language processing techniques such as stemming, lemmatizing, and removal of stop words. We used Python's VADER library to identify and classify the sentiment (positive, negative, or neutral) and subjectivity (objective or subjective) of tweets [14]. VADER applies a rule-based sentiment analysis with a polarity scale of −1 (most negative) to 1 (most positive).

For the subjectivity analysis, we used TextBlob to label each tweet from a range of 0 (objective) to 1 (subjective). Objective tweets relay facts, whereas subjective tweets typically communicate an opinion or belief. For the two hashtags #plandemic and #scamdemic, we visualized sentiment using a histogram of the subjectivity scores.

We used the Python library NRCLex to label the primary emotion for each tweet (fear, anger, anticipation, trust, surprise, positive, negative, sadness, disgust, or joy) [15].

### Topic modeling

To identify the major topics discussed in our tweet library, we used the Gensim library in Python and applied an unsupervised machine-learning algorithm called Latent Dirichlet Allocation (LDA), which identifies clusters of tweets by a representative set of words [16].

We used the most highly weighted words in each cluster to determine the content of each topic. To find the optimal number of topics required by LDA, we trained several LDA models using different numbers of topics ranging from 2 to 100 and computed a topic coherence score (produced by evaluating the relative distance between the topics' most highly weighted words) for each LDA model. We ultimately chose a twelve-topic LDA model as it maximized the coherence score. One author without access or insight into the topic model labeled the topics using the 30 most frequently used terms ordered by weight. All authors then evaluated these topic labels and reached a consensus.

### Demographics

Using m3 inference, we obtained user demographics including gender, age group, and type of account [17]. To obtain the ethnicity, we used the ethnicolr library in Python to predict the ethnicity of the user [18].

### Results

We identified 420,107 tweets in 2020 that contained the keywords #scamdemic and #plandemic. After removal of tweets that were replies, retweets, non-English tweets, or duplicate tweets, we retained 227,067 tweets from 40,081 users. Fig 1 shows a word cloud of common words used in tweets with size denoting frequency of use.

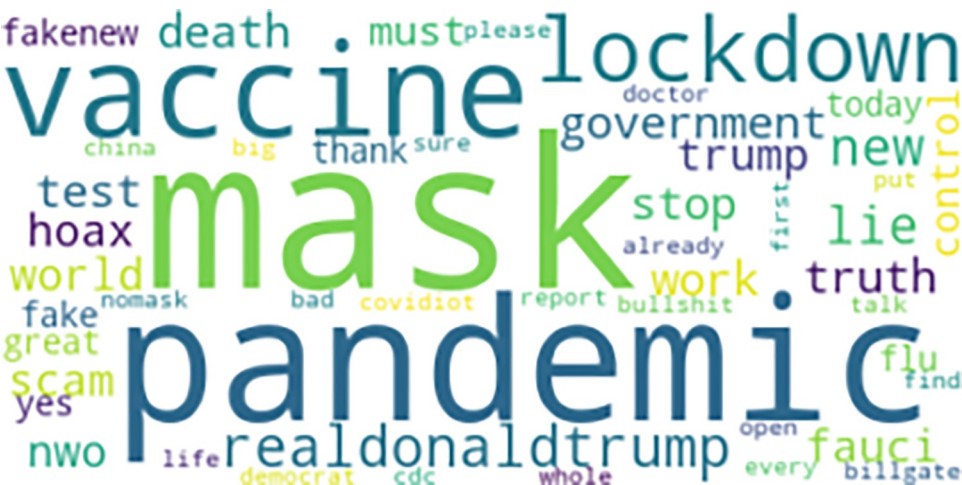

**Fig 1. Word cloud of common words used in tweets.**

### Descriptive analysis

Of 227,067 total tweets, 168,836 (74.4%) tweets were published by 31,405 (78.4%) active users (5.4 tweets per user) and 58,231 (25.6%) were by 8,676 (21.6%) users (6.7 tweets per user), whose account had been suspended by January 15, 2021. Users who were suspended were statistically more likely to tweet more (p = 0.004) and users who used both hashtags were more likely to be suspended (29.2%) than those that used #plandemic (25.9%) or #scamdemic (13.2%) only. Of tweets with both hashtags, 11,174 (28.3%) tweets were suspended compared to 37,454 (34.7%) of #plandemic and 9,603 (120.0%) of #scamdemic tweets.

Twitter Web App was the most used platform by active (32.6%) and suspended (31.4%) users followed by Twitter for iPhone (28.2% and 290.0%). Less than 20% of tweets had media (image or video) and about one-quarter of tweets contained a URL. The median active user had over 8,000 posts and 470 followers and the median suspended user had over 12,000 posts and 610 followers. None of the users who tweeted the selected hashtags had his/her identity verified ✅ (blue checkmark) by Twitter. Table 1 shows the demographics of twitter users including age, gender, and ethnicity. Non-Hispanic Black users were significantly more likely to be suspended than active (11.3% vs 9.7%, $P < 0.001$) whereas Hispanic users were significantly less likely to be suspended (3.2% vs 5.1%, $P < 0.001$).

The largest group of users were 40 years or older. Males and non-Hispanic Whites represented the largest groups. (Table 1) Male users and users in the age groups < = 18 years and 30–39 years were overrepresented significantly among the suspended users. The vast majority of active and suspended users tweeted from personal accounts, 88.2% and 79.4% respectively.

We listed the characteristics of tweets in Table 2. Among all tweets, suspended tweets were significantly more likely to have likes ($P < 0.001$) and retweets ($P < 0.001$) compared to active tweets. The average number of hashtags per tweet was three (range 1–5), except active accounts using #scamdemic had an average of two per tweet.

### Objectivity/Subjectivity analysis

On a scale from 0 (objective) to 1 (subjective), the set of tweets were primarily more objective in nature with 65% demonstrating near or complete objectivity (Fig 2). The median subjectivity score for #plandemic was 0.22 (interquartile range [IQR], 0–0.45) and 0.22 for #scamdemic (interquartile range [IQR], 0–0.46) (Table 3).

**Table 1. Characteristics of Twitter users.**

|  |  | **Active Users** | **Suspended Users** |
|---|---|---|---|
| **User Characteristics** |  | *n = 31,405* | *n = 8,676* |
| Verified twitter account |  | 0 | 0 |
| User followers |  | 472 (118–1565) | 610 (165–1907) |
| User posts to date |  | 8,394 (2,680–24,677) | 12,786 (3,668–26,023) |
| **Demographic Characteristics** |  |  |  |
| Age Group: |  | *n = 31,405 (78.4%)* | *n = 8,676 (21.6%)* |
|  | < = 18 | 3,611 (11.5)* | 2,178 (25.1) |
|  | 19–29 | 7,129 (22.7)* | 902 (10.4) |
|  | 30–39 | 6,501 (20.7)* | 2,473 (28.5) |
|  | > = 40 | 14,164 (45.1)* | 3,123 (36.0) |
| Gender: |  |  |  |
|  | Male | 22,580 (71.9)* | 7,002 (80.7) |
|  | Female | 8,825 (28.1)* | 1,674 (19.3) |
| Ethnicity: |  |  |  |
|  | Asian | 3,078 (9.8) | 819 (9.44) |
|  | Hispanic | 1,602 (5.1)* | 281 (3.23) |
|  | Non-Hispanic Black | 3,046 (9.7)* | 983 (11.33) |
|  | Non-Hispanic White | 23,679 (75.4) | 6,593 (76.0) |
| Type of Account: |  |  |  |
|  | Organization | 3,392 (10.8)* | 816 (9.4) |
|  | Personal | 28,013(89.2)* | 7,860 (90.6) |

* = Significant difference between active and suspended users ($P < 0.001$).

Comparison between the two groups is done using Welch's t-test.

## Emotion analysis

In the analysis of emotions expressed in the tweets, fear was the most common emotion followed by trust, sadness, and anger. Disgust, surprise, and joy were least expressed (Fig 3). Suspended tweets were statistically more likely to express anger, disgust, and surprise.

## Sentiment analysis

The overall sentiment for #plandemic and #scamdemic was negative, as noted in Fig 3. The mean weekly sentiments for #plandemic and #scamdemic were negative throughout the study period (Fig 4) with an overall mean sentiment -0.05 and -0.09 for #plandemic and #scamdemic, respectively (-1 denotes completely negative, 1 completely positive). During the week of May 4th, 2020, the movie *Plandemic: Indoctornation [19]* was released, after which the polarity for both hashtags became more negative for several weeks. During the week of the United States election, there was a slight uptick in the mean polarity towards neutral, but following the election, the mean polarity became more negative for both hashtags, and for the first time, the mean polarity of #plandemic was more negative than #scamdemic.

## Topic modeling

LDA identified 12 topics in our tweet collection and we subjectively labeled them based on the predominant keywords. (Table 4) The content of tweets were almost exclusively (>99%) representative of a single topic. The most frequent tweet topic was "Complaints against mandates introduced during the pandemic" (79,670 tweets), which included complaints against masks,

**Table 2. Characteristics of tweets.**

| | All Tweets | | #plandemic | | #scamdemic | | #plandemic AND #scamdemic | |
|---|---|---|---|---|---|---|---|---|
| | *Active* | *Suspended* | *Active* | *Suspended* | *Active* | *Suspended* | *Active* | *Suspended* |
| Tweets<br>Users | n = 168,836[†]<br>n = 31,405 | n = 58,231<br>n = 8676 | n = 70,436[†]<br>n = 23,534 | n = 37,454<br>n = 8,229 | n = 70,116[†]<br>n = 11,788 | n = 9,603<br>n = 1,791 | n = 28,284[†]<br>n = 3,552 | n = 11,174<br>n = 1,468 |
| **Characteristics:** | | | | | | | | |
| Has link | 38,573[†] (22.8) | 11,163 (19.2) | 18,283[†] (260.0) | 6,669* (17.8) | 12,366[†] (17.6) | 2,176 (22.7) | 7,888[†] (27.9) | 2,318 (20.7) |
| Mentions | 19,266[†††] (11.4) | 6,912 (11.9) | 9,278 (13.2) | 4,832* (12.9) | 7,736 (110.0) | 1,018 (10.6) | 2,252[†] (80.0) | 1,062 (9.5) |
| Has media | 30,244[†] (17.9) | 11,251 (19.3) | 13,400 (190.0) | 6,976* (18.6) | 10,687[†] (15.2) | 1,686 (17.6) | 6,157[†††] (21.8) | 2,589 (23.2) |
| Has reply | 31,706 (18.8) | 10,770 (18.5) | 12,969[†] (18.4) | 6,583* (17.6) | 13,563[†] (19.3) | 2,152 (22.4) | 5,174 (18.3) | 2,035 (18.2) |
| Has like | 74,840[†] (44.3) | 27,305 (46.9) | 30,287[†] (430.0) | 16,82* (44.9) | 31,530[†] (450.0) | 4,875 (50.8) | 13,023[†] (460.0) | 5,609 (50.2) |
| Has retweet | 38,596[†] (22.9) | 17,266 (29.7) | 16,683[†] (23.7) | 10,677* (28.5) | 13,996[†] (200.0) | 3,068 (31.9) | 7,917[†] (280.0) | 3,521 (31.5) |
| **Twitter Source:** | | | | | | | | |
| Twitter for iPhone | 47,612[†] (28.2) | 16,886 (290.0) | 20,919 (29.7) | 11,086* (29.6) | 18,931 (270.0) | 2,679 (27.9) | 7,778 (27.5) | 3,106 (27.8) |
| Twitter for Android | 34,105 (20.2) | 11,763 (20.2) | 13,876 (19.7) | 7,453** (19.9) | 14,304 (20.4) | 1,940 (20.2) | 5,883 (20.8) | 2,380 (21.3) |
| Twitter Web App | 55,041[†] (32.6) | 18,285 (31.4) | 21,483 (30.5) | 11,348* (30.3) | 23,839 (340.0) | 3,236 (33.7) | 9,701 (34.3) | 3,721 (33.3) |
| Instagram | 1,182[†] (0.7) | 524 (0.9) | 563 (0.8) | 337 (0.9) | 421[††] (0.6) | 77 (0.8) | 226 (0.8) | 89 (0.8) |

Note

\* = Significant difference of Suspended tweets between the hashtag groups ($P < 0.001$).

\*\* = Significant difference of Suspended tweets between the hashtag groups ($P = 0.005$).

† = Significant difference from Active and Suspended tweets ($P < 0.001$).

†† = Significant difference from Active and Suspended tweets ($P = 0.02$).

††† = Significant difference from Active and Suspended tweets ($P = 0.003$).

Comparison between the two hashtags and tweet status performed using chi-square testing.

social distancing, and closures, and had the highest percentage of suspended tweets. The next most popular topics included tweets "downplaying the dangers of COVID-19" (23,185 tweets), "Lies and brainwashing by the media and politicians" (18,871 tweets), and "Corporations and global agenda" (15,493 tweets). Overall topics had tweet suspension rates ranging from 16.6% to 36% (Table 4).

## Discussion

Social Media can be the source of Misinformation, Disinformation, and Malinformation. We analyzed two hashtags that represent mis- and disinformation (#plandemic and #scamdemic) to analyze the extend of false information in social media.

### Suspended tweets and users

Our observations of tweets for the year 2020 showed that more than 1 in 5 Twitter users (21.6%), who used any of the hashtags #plandemic or #scamdemic during 2020 had their

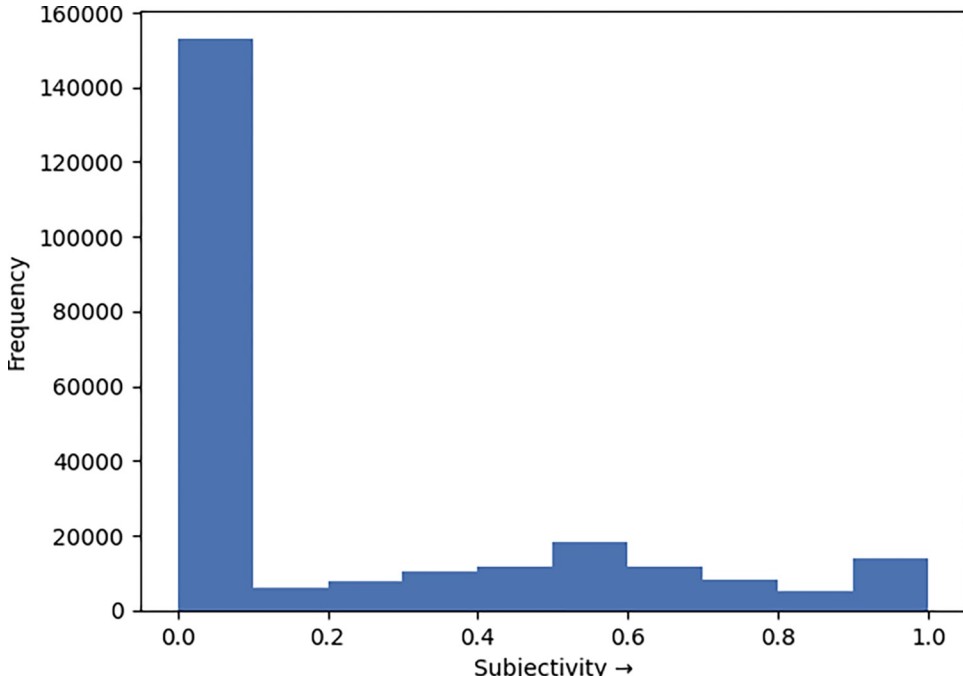

**Fig 2. Objectivity/Subjectivity analysis of tweets.** 0 represents complete objectivity, 1 represents complete subjectivity.

accounts suspended in 2020. Suspended users were disproportionately more likely to be less than 18 years old or between 30 and 39 years old. Even though women use twitter more actively [20], men were more likely to use the selected hashtags in the first place and they were significantly overrepresented among the suspended users, which may reflect the fact that men are more likely to use taboo words or topics in tweets [21]. Accounts by non-Hispanic blacks and private individuals (vs. organizations) were disproportionally suspended.

Twitter suspensions have been historically linked to politics as a major theme, as with our hashtags [22]. Suspended tweets were statistically more likely to have likes, media content, and retweets and they were less likely to have links or mentions. The last finding that suspended tweets had less links (e.g., to newspaper articles) or mentions suggests that the tweets were less likely to report a verifiable fact than could be validated by readers. Suspended tweets were more likely to be engaging as indicated by a significantly higher rate of likes and retweets; however, this finding may also be attributable to previously reported communities that spread

**Table 3. Example tweets with subjectivity and sentiment scores for each hashtag.**

|  | Number of Tweets | Mean Sentiment | Mean Subjectivity | Representative Tweet |
|---|---|---|---|---|
| #plandemic | 82,739 | -00.001 | 0 | Total US Deaths 2018 2.84 million Total US Deaths 2019 2.85 million Total US Deaths 2020 as of December 4th 2,654,825 (Pulled from CDC Site) Yea we've all been duped. . . #plandemic #FauciFraud #EndLockdowns |
| #plandemic | 5,448 | -0.198 | 1 | #plandemic What a bunch of nonsense over what is basically a cold. |
| #scamdemic | 64,681 | -00.001 | 0 | Have you seen this? As of 19 March 2020, COVID-19 is no longer considered to be a high consequence infectious diseases (HCID) in the UK. #scamdemic #WeHaveBeenHad #WhoWillPay |
| #scamdemic | 4,343 | -0.208 | 1 | Just as "CO2-based climate change" is a massive worldwide hoax, so is the #covid19 fake "pandemic". And the same people are pushing these hoaxes, for the same agenda: totalitarian government control. #Event202 #scamdemic #plandemic #MedicalMartialLaw |

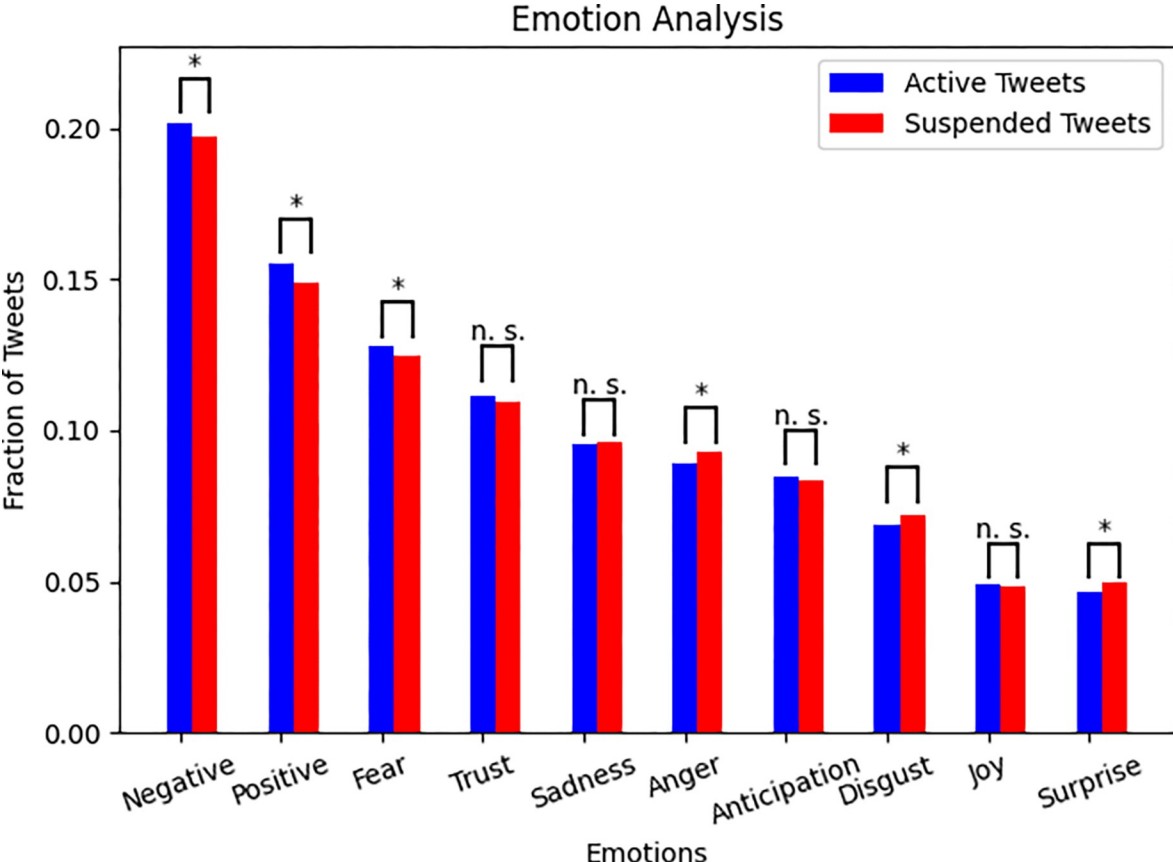

**Fig 3. Distribution of predominant emotions and valence in tweets by active and suspended tweets.** Note: * = Significant difference where $P < 0.001$ for Negative and Positive, P = 0.01 for Fear and Disgust, P = 0.004 for Anger, and P = 0.001 for Surprise. n. s. = Not significant.

misinformation [21]. As suspension on Twitter usually is triggered through crowdsourcing of users who report offensive or problematic tweets, tweets with more likes and shares that add to their distribution are more likely to be suspended.

## Emotion analysis

The emotions fear, sadness, anger, and disgust were more frequently expressed than joy and surprise. Tweets that expressed emotions linked to fight-or-flight responses such as anger, disgust, and surprise were more likely to be suspended–perhaps because they triggered stronger emotions in readers resulting more reporting activity.

## Objectivity & sentiment

The Objectivity/Subjectivity analysis of the tweets showed a predominance of subjective tweets. However, we realized many tweets in our collection were labeled by our tool as objective while the actual meaning was sarcastic. Sarcasm is a sophisticated construct to express contempt or ridicule. Tweets with sarcasm are thus rather subjective in nature [23]. Sarcasm has been shown to be the main reason behind false classification of tweets [24].

Phrasing a tweet in an objective manner does not mean that the content of the tweet is true. While 65% of tweets were labeled as purely objective in nature, they contained mis- and disinformation that was expressed in an objective fashion.

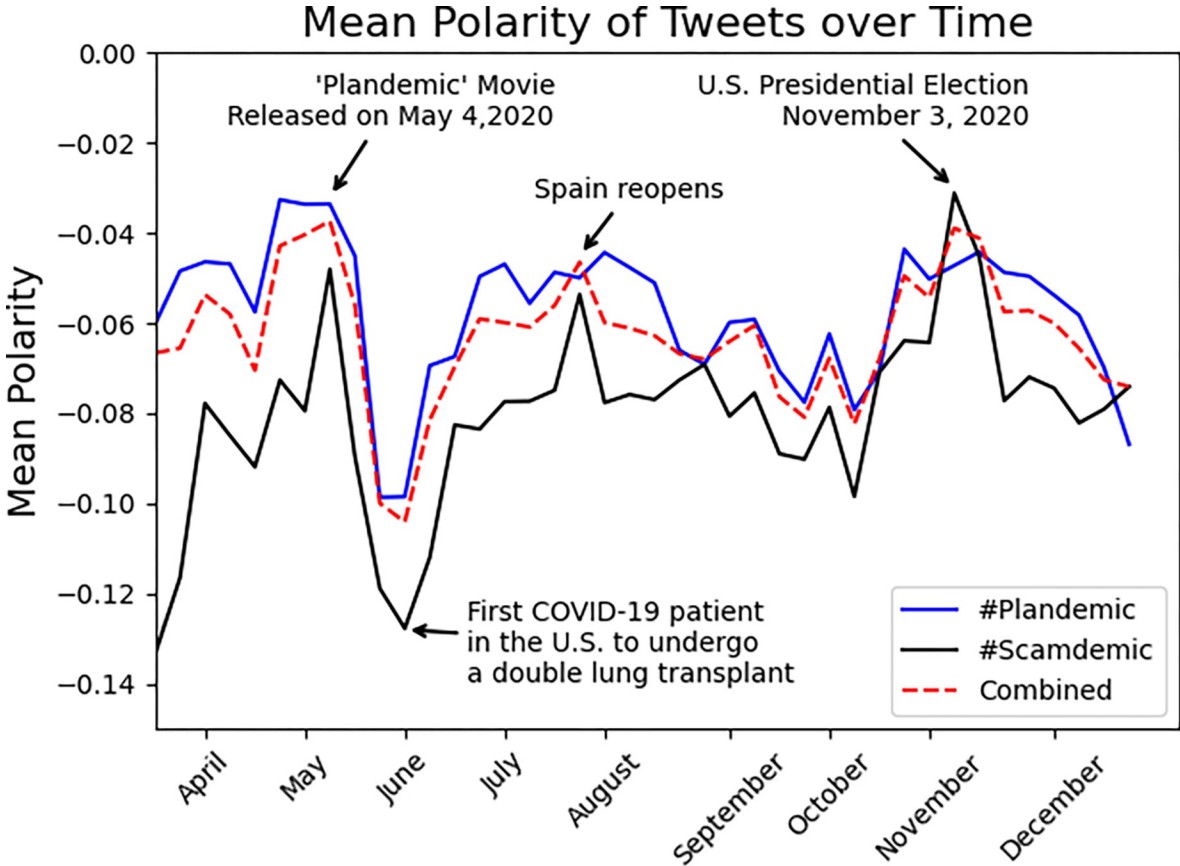

**Fig 4. Mean polarity of #scamdemic and #plandemic tweets over time.**

Unlike our prior study looking at general COVID-19 related tweets [4], where we found a predominantly positive sentiment, the mean sentiments of the tweets in this study were expectedly more negative. Media events like the release of the 'Plandemic' movie further negatively affected sentiments.

## Topic modeling

Our machine learning approach derived 12 main topics. Three topics were closely related, dealing with anger of pandemic mandates (shutdowns, masks, etc.) and politicians. Two topics focused on the roles of the media and corporations. Another four topics focused on downplaying the dangers of COVID-19 or the pandemic being a hoax or exaggerated. One standalone topic focused on the censoring of COVID-19 deniers and two advertised "documentaries" on COVID-19 or distributed vaccine misinformation.

## Suspensions

Our analysis of tweets in 2020 with the hashtags #scamdemic or #plandemic provides important insight into the disinformation distributed on Twitter. One surprising finding was the rate by which users, who used the hashtags were suspended by Twitter. One fifth, who used the hashtags, had a suspension of their accounts by January 2021. Twitter allows users to report misleading tweets and to categorize them as health related and COVID-19 related tweets.

**Table 4. Tweet topics, frequency, most common keywords, and sample tweets.**

| Possible Topic Label | Tweets/ Topic | Suspended Tweets (%) | Words Contributing to Topic Model (in Decreasing Order of Weight) | Representative Tweet |
|---|---|---|---|---|
| **Complaints against Pandemic Mandates** | 79,670 | 28,670 (35.99) | know, get, stop, realdonaldtrump, keep, mask, scam, back, go, people, need, fuck, let, want, think, would, see, one, way, like | I highly doubt the cloth mask wearers are washing them daily. Gross! Just wash your hands. Don't touch your face. Stay home when sick. It's all very basic. I've been doing those things long before Covid! #scamdemic |
| **Downplaying the danger** | 23,185 | 4,649 (20.05) | pandemic, death, people, test, plan, case, flu, number, end, many, would, vaccine, die, say, lockdown, control, get, yet, see, year | Someone please explain to my why there is an alleged 1% chance of having a fatal reaction to a vaccine, but we don't completely shut down the practice of vaccinating? You have a greater chance of dying from vaccines than you do from COVID-19. #plandemic |
| **Lies & Brainwashing by media and politicians** | 18,871 | 3,788 (20.07) | people, lie, right, like, see, think, make, go, happen, start, believe, bad, look, say, remember, get, nothing, shit, many, mask | The fact you have to ASK for cases just shows what bullshit this Covid crap is. Channel 7 doing the same thing with these so call "frontline nurses" playing their part (badly aswell). Wake up and focus on the real issues and stop placing fear in people. #FakeNews #scamdemic https://t.co/TsCKv1HC3f |
| **Corporations and the globalist agenda** | 15,493 | 3,027 (19.54) | people, want, take, see, government, get, vaccine, know, make, would, go, need, money, say, fact, fear, leave, really, billgate, push | The propaganda around the #plandemic is ramped up as Senate hearing is being used to push flu shots and vaccines to fearful member of public. . . .be weary of anyone trying to medicate you and convince you to put man made substances into your body. . . That They Have Made |
| **Anger over restrictions and politicians** | 14,923 | 2,976 (19.94) | state, use, get, protest, control, mask, kid, people, call, school, governor, child, like, support, allow, make, tyranny, america, wake, wow | Stand up, fight back and remove every Governor and Mayor who is not upholding the Constitution. #FightBack #plandemic #OpenAmerica #FreedomIsntFree #Constitution #SayNoToCommunism #insurgency #coup #SaveOurChildren #PatriotsUnited #NWO #Agenda2021 #depopulation #freedomoverfear |
| **Censuring COVID deniers** | 14,807 | 2,977 (20.11) | one, day, say, need, people, go, speak, take, doctor, part, work, tell, lockdown, come, think, proof, another, tweet, many, actually | Notice how many health professionals who speak out get censored? Doesn't surprise me that more can't speak out. #scamdemic2020 #scamdemic #plandemic #Covid_19 #COVID #COVIDIOTS #COVID19 #COVIDSecondWave |
| **Vaccine misinformation** | 14,397 | 2,915 (20.25) | go, watch, vaccine, say, get, even, tell, know, work, everyone, new, cure, still, people, call, lol, wait, see, make, year | #modernavaccine 93% effective #Pfizervaccine 95% effective #ImmuneSystem 99.99% effective against #COVID19 Why do we need a vaccine again? Where is all the death? #CovidHoax #plandemic #Agenda2030 |
| **Vote the lying politicians out** | 14,378 | 3,126 (21.74) | time, real, people, fake, never, get, plandemic, realdonaldtrump, please, exactly, well, like, country, say, wake, need, really, do, vote, open | https://t.co/QNQuzGzach Thx Governor Doosh Bag for falling for the fake plandemic. @dougducey if you would just have kept the economy open you would not be killing more people from Eviction/Joblessness than the Virus killed. . . . #plandemic #Economic #Collapse #Resign #Govenor |
| **The pandemic is exaggerated** | 11,749 | 2,235 (19.02) | would, like, expose, video, fraud, get, agenda, say, man, vaccine, world, another, hospital, test, order, one, also, work, truth, make | All we keep hearing is "stay at home so we don't overwhelm the NHS". . . .. look at all the NHS staff who have hours of spare time to tweet, we've all seen the tiktok dances and the empty hospitals. . . .. Instead of telling us, why don't they show us hospitals OVERWHELMED? #plandemic |
| **Plandemic Documentaries** | 8,755 | 1,451 (16.57) | yes, joebiden, true, bullshit, read, article, share, fauci, see, china, people, play, video, today, create, mask, make, documentary, tell, facebook | @TuckerCarlson This Is CNN: Oliver Darcy Gets Facebook to Pull James O & Keefe Coronavirus Video Report https://t.co/nRYbKhJF2h tm_campaign = websitesharingbuttons The media diagnosed the virus and giving the prescription then involved in taking away our rights.#scamdemic |
| **COVID is a hoax** | 6,744 | 1,350 (20.02) | hoax, lockdown, science, mask, die, sweden, think, due, death, say, need, drjudyamikovit, thought, people, brilliant, les, number, stand, corruption, sign | The hoax is over. End the lockdowns NOW. #scamdemic #TheGreatReset #plandemic #COVID #HOAX |
| **Canadian Virus Hoax** | 5,517 | 971 (17.60) | con, truth, come, por, los, van, una, victim, check, est, dat, canada, een, ready, trump, niet, las, nope, dan, alert | Public Health Canada admits to overinflating death rates by 50% https://t.co/cH4JLKquxG #StepUpSingh #StepUpNDP #OpenTheWorld #EndTheLockdown #MasksOff #MasksDontWork #Scamdemic #NoVaccine #CoronavirusFacts #CdnPoli https://t.co/g93dIQMOCz |

## Limitations

Our study was limited by several factors. First, we selected a subset of tweets designed to provide us with tweets containing disinformation. As such, our library of tweets contained many tweets including sarcasm, which limited our ability to use tools we had used in prior studies [4,7]. Second, we used existing tools to analyze sentiments and emotion of tweets that are not specific to health care topics, which could have skewed our analysis. Finally, since we targeted only tweets in English and are unable to determine geographic location for users, we are limited in making conclusions about specific countries or countries where English is the not the predominant language.

## Potential interventions

Our study demonstrates that it is possible to identify disinformation from tweets. In the future, public health agencies could automate the tools used to identify disinformation in real time and target it with replies that disseminate correct but related educational information. We envision public health "bots" as a means of de-arming disinformation spreaders.

## Conclusions

Leveraging 227,067 tweets with the hashtags #scamdemic and #plandemic in 2020, we were able to explore topics successfully, and user demographics to elucidate important trends in public disinformation about the COVID-19 vaccine. In general, COVID-19 tweets demonstrated overall negative sentiment. Besides expressing anger over pandemic restrictions, substantial amounts of tweets were dedicated to presenting disinformation. More than one in five users who used these hashtags in 2020, were suspended by Twitter in January 2021.

## Supporting information

**S1 File. This is the S1 File title.**
(DOCX)

**S2 File. This is the S2 File title.**
(CSV)

## Acknowledgments

Declarations

### Ethics approval and consent to participate

The University of Texas Southwestern Human Research Protection Program Policies, Procedures, and Guidance did not require institutional review board approval as all data were publicly available.

## Author Contributions

**Conceptualization:** Christoph U. Lehmann, Richard J. Medford.

**Data curation:** Heather D. Lanier, Marlon I. Diaz.

**Formal analysis:** Marlon I. Diaz.

**Investigation:** Heather D. Lanier.

**Methodology:** Heather D. Lanier, Marlon I. Diaz, Sameh N. Saleh, Christoph U. Lehmann, Richard J. Medford.

**Supervision:** Christoph U. Lehmann.

**Visualization:** Heather D. Lanier.

**Writing – original draft:** Heather D. Lanier, Marlon I. Diaz, Christoph U. Lehmann, Richard J. Medford.

**Writing – review & editing:** Sameh N. Saleh, Christoph U. Lehmann, Richard J. Medford.

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
