## [Decision Letter · Decision Letter 0]

30 Dec 2021

PONE-D-21-33728Analyzing COVID-19 Disinformation on Twitter using the Hashtags #scamdemic and #plandemic: Retrospective StudyPLOS ONE

Dear Dr. Lanier,

Thank you for submitting your manuscript to PLOS ONE. After careful consideration, we feel that it has merit but does not fully meet PLOS ONE’s publication criteria as it currently stands. Therefore, we invite you to submit a revised version of the manuscript that addresses the points raised during the review process. Please follow recommendations from Reviewers.

We look forward to receiving your revised manuscript.

Kind regards,

Jarosław Jankowski

Academic Editor

PLOS ONE

Journal Requirements:

Reviewers' comments:

Reviewer's Responses to Questions

**Comments to the Author**

1. Is the manuscript technically sound, and do the data support the conclusions?

Reviewer #1: Yes

Reviewer #2: Yes

Reviewer #3: Yes

2. Has the statistical analysis been performed appropriately and rigorously? 

Reviewer #1: Yes

Reviewer #2: I Don't Know

Reviewer #3: Yes

3. Have the authors made all data underlying the findings in their manuscript fully available?

Reviewer #1: Yes

Reviewer #2: Yes

Reviewer #3: No

4. Is the manuscript presented in an intelligible fashion and written in standard English?

Reviewer #1: Yes

Reviewer #2: Yes

Reviewer #3: Yes

5. Review Comments to the Author

Reviewer #1: Useful research, it would be nice to extend the conclusions with some results' implications part - with one targeting social implications - educational ones. Even without this implications part, conclusions might be more elaborated, they have been synthesized to a minimum.

Reviewer #2: INTRODUCTION

• The literature review should be precisely and concisely presented.

• The gap of knowledge and necessity of the study should be precisely stated.

• Coherence within paragraph and between paragraphs and sentence sequence were not well designed.

• What is statement “Error! Bookmark not defined” in line78

METHODS

• The statistical analysis should be precisely stated.

• Time of data collection should be clarified and be the same in all parts of manuscript.

• Some result of data collection mention in method section l96-98. Please move to result section.

RESULTS AND DISSCUSSION

• Some percentage in the result’s text and tables were over 100%. please revise them.

• It is not necessary to bring all the table’s contents in the text, just mention main results.

• Discussion is weak and repeats the results, please rewrite again or improve them.

Reviewer #3: The authors analyze tweets with the hashtags #scamdemic and #plandemic in 2020 to provide insights into disinformation about the COVID-19 pandemic. The analysis is suitable for answering the question. I support this paper to be published.

Here are some questions I’m wondering:

1. In your emotion analysis, the authors label the ten emotions for each tweet: fear, anger, anticipation, trust, surprise, positive, negative, sadness, disgust, or joy. These are not common basic emotions. Is there theoretical support for the choice of these emotions?

2. At the beginning of the discussion, the authors describe different types of false information in social media. Isn’t it better to be described in the introduction?

3. The authors mention this paper can aid in developing targeted public health interventions. Could the authors write down several paragraphs about what kind of interventions can be developed to tackle the disinformation about the COVID-19 pandemic based on the findings? Adding this implication part in the discussion may add more value to your paper.

6. PLOS authors have the option to publish the peer review history of their article (what does this mean?). If published, this will include your full peer review and any attached files.

Reviewer #1: No

Reviewer #2: **Yes: **Farideh Mostafavi

Reviewer #3: No

---

## [Author Response · Author response to Decision Letter 0]

18 Mar 2022

Responses to comments have been attached as a separate file

---

## [Decision Letter · Decision Letter 1]

29 Apr 2022

Analyzing COVID-19 Disinformation on Twitter using the Hashtags #scamdemic and #plandemic: Retrospective Study

PONE-D-21-33728R1

Dear Dr. Lanier,

We’re pleased to inform you that your manuscript has been judged scientifically suitable for publication and will be formally accepted for publication once it meets all outstanding technical requirements.

Kind regards,

Daswin De Silva

Academic Editor

PLOS ONE

Additional Editor Comments (optional):

Reviewers' comments:

Reviewer's Responses to Questions

**Comments to the Author**

1. If the authors have adequately addressed your comments raised in a previous round of review and you feel that this manuscript is now acceptable for publication, you may indicate that here to bypass the “Comments to the Author” section, enter your conflict of interest statement in the “Confidential to Editor” section, and submit your "Accept" recommendation.

Reviewer #1: All comments have been addressed

Reviewer #2: All comments have been addressed

2. Is the manuscript technically sound, and do the data support the conclusions?

Reviewer #1: Yes

Reviewer #2: Yes

3. Has the statistical analysis been performed appropriately and rigorously? 

Reviewer #1: Yes

Reviewer #2: Yes

4. Have the authors made all data underlying the findings in their manuscript fully available?

Reviewer #1: Yes

Reviewer #2: Yes

5. Is the manuscript presented in an intelligible fashion and written in standard English?

Reviewer #1: Yes

Reviewer #2: Yes

6. Review Comments to the Author

Reviewer #1: The article is in a better form, suggestions have been considered. The research subject is important and the research method quite interesting. For future researches it would be also nice to look for potential explanations for the emotions discovered in tweets.

Reviewer #2: The statement “Error! Bookmark not defined” in line78 is not removed yet.

Please move or state statistical analysis in the main text too.

7. PLOS authors have the option to publish the peer review history of their article (what does this mean?). If published, this will include your full peer review and any attached files.

Reviewer #1: No

Reviewer #2: **Yes: **Farideh Mostafavi

---

## [Editor Report · Acceptance letter]

30 May 2022

PONE-D-21-33728R1 

Analyzing COVID-19 Disinformation on Twitter using the Hashtags #scamdemic and #plandemic: Retrospective Study 

Dear Dr. Lanier:

I'm pleased to inform you that your manuscript has been deemed suitable for publication in PLOS ONE. Congratulations! Your manuscript is now with our production department. 

Kind regards, 

on behalf of

Dr. Daswin De Silva 

Academic Editor

PLOS ONE